

# Effects of mucus trail following on the distance between individuals of opposite sex and its influence on the evolution of the trait in the Ezo abalone *Haliotis discus hannai*

Yukio Matsumoto

Miyako Laboratory, Tohoku National Fisheries Research Institute, Japan Fisheries Research and Education Agency, Miyako, Iwate, Japan

## ABSTRACT

**Background**. Aggregation affects the fertilization rate of species that utilize external fertilization. However, the process of aggregation has not been studied in detail, using either theoretical models or real world observations. We used the Ezo abalone *Haliotis discus hannai* as a model animal species to evaluate whether mucus trail following (MTF) facilitates aggregation by reducing the distance between members of opposite sexes. We also examined whether the fertilization rate increase with mucus-trail-following is an evolutionary driving force in mucus following behavior.

**Methods**. We used a y-maze to test whether *H. discus hannai* follows the mucus trails of other individuals. Distances between members of the opposite sex of MTF individuals were compared to non-MTF individuals using an individual-based model (IBM) consistent with the behavior of *H. discus hannai*. To examine whether MTF behavior evolved to reduce distances between members of the opposite sex, we constructed simple population genetic models of a diploid population with nonoverlapping, discrete generations.

**Results**. *Haliotis discus hannai* chose the y-maze arm with the mucus trail more frequently than the one without, regardless of the sex of the abalone that secreted the mucus or the reproductive season. In the IBM the distance between opposite sexes was significantly reduced by MTF behavior; however, the difference in distances between opposite sex compared to same sex individuals was only several centimeters. Simple population genetic models indicated that the aggregating effect of MTF between the opposite sex members could be an evolutionary driving force.

**Conclusions**. These results suggest that observed MTF behavior might have evolved as a mechanism to increase the fertilization rates although other factors could also be involved.

Corresponding author
Yukio Matsumoto,
yukio.matsumoto@live.jp

## INTRODUCTION

Studies of the Allee effect in externally fertilized species are important for population management because a decreasing density of adults decreases the fertilization rate, potentially to the point of population collapse (*Berec, Angulo & Courchamp, 2007*). Previous studies indicated that water flow (*Babcock & Keesing, 1999*), spawning synchrony (*Calabrese & Fagan, 2004*), and the distance between opposite sexes (*Babcock & Keesing, 1999*; *Levitan, Sewell & Chia, 2016*) affect egg fertilization. Although the distance between individuals of opposite sex decreases naturally with increasing population density, distance can also decrease because of aggregation behavior. Theoretical models including aggregation behavior demonstrate higher rates of fertilization compared to those without at the same population density (*Claereboudt, 1999*; *Lundquist & Botsford, 2004*; *Zhang, 2008*). However, the mechanism underlying aggregation has not been widely studied across multiple species or through theoretical models. The cues used by individuals to recognize the opposite sex will affect the distance over which they can respond. For example, when individuals visually recognize others the encounter rate is low at low population densities; whereas if individuals recognize each other through olfaction they can search for the opposite sex from a distant location (*Jumper & Baird, 1991*). Although a model has been developed that considers the distance over which individuals can recognize the opposite sex (*Baird & Jumper, 1995*; *Coates & Hovel, 2014*), the cues used by animals for recognition remains unclear.

It has been reported that gastropod species follow mucus trails for mate searching, as they prefer to follow mucus from the same species (*Nakashima, 1995*) and from the opposite sex (*Erlandsson & Kostylev, 1995*; *Johannesson et al., 2010*; *Ng et al., 2011*). However, the extent to which the efficiency of mate searching is increased by mucus trail following (MTF) has not been tested. In addition, it is noteworthy that the reproductive success of individuals that do not follow mucus trails could increase when they are followed by the mucus-following individuals that follow the mucus. This phenomenon should be considered in the evolutionary process of MTF behavior. In the earlier stages of a trait's evolution it is generally assumed that a small number of individuals with the trait (i.e., mutant type) occur within a population without the trait (wild type): when the mutant type leaves relatively more offspring than the wild type its frequency should increase. When applying this conceptual framework to the evolution of MTF, it is assumed animals that follow mucus trails (i.e., mutant type) and those that do not (i.e., wild type) have coexisted. In this situation, the reproductive success of the wild type could be increased when the mutant type follows the mucus of the wild type. In other words, as the difference in fitness between mutant type and wild type might be small, the greater efficiency in mate searching might not be the evolutionary driving force of MTF behavior.

In *Haliotis* species, which uses external fertilization, aggregation and distance between opposite sexes have been investigated as the index indicating the encounter of opposite sex members (e.g., *Button, 2008*). The current study was conducted to test the relationship between MTF and distance between opposite sexes using *H. discus hannai* as the model animal. First, we confirmed that *H. discus hannai* exhibits non-self MTF behavior. Next,

we tested whether MTF reduces the distance between opposite sexes. One way to examine the effect of MTF on mate searching is by comparing the distances between opposite sex individuals that exhibit MTF and those that do not. Non-mucus following individuals were obtained by excision of the cephalic tentacle, the sensory organ that detects mucus in *Haliotis* species (*Kuanpradit et al., 2012*). However, it is highly possible that such an invasive procedure causes subsequent abnormal behavior. We therefore created an Individual Based Model (IBM) in which some individuals exhibited MTF behavior and some did not. We analysed this model to test whether reduced distances between opposite sexes is an evolutionary driving force of MTF behavior.

## MATERIALS & METHODS

### Study species

*Haliotis discus* species is a marine gastropod mollusk in the family Haliotidae, which is distributed in the waters off Japan and eastern Asia. The species mainly inhabits rocky shores. The spawning season in the Iwate Prefecture, Japan, is from August to late October, when water temperatures are around 20–21 °C after the effective accumulated temperature (EAT; *Uki & Kikuchi, 1984*) reaches 1000 degree-days. Experimental animal subjects were captured in the ocean of Iwate Prefecture in October 2015 and segregated by gender in tanks equipped with water temperature control.

### Mucus trail following

To evaluate whether *H. discus hannai* follows mucus trails, binary choice tests were conducted using a y-maze, taking place on December 11–28, 2015, August 17–20, 2016, and June 26-July 17, 2017. In this study, individuals with immature gonads were classed as non-reproductive individuals (experiments conducted during December 11–28, 2015) and individuals at simulated EAT 1000–2000 degree-days were classed as reproductive individuals (experiments conducted during August 17–20, 2016, and June 26–July 17, 2017). The y-maze consisted of a 50-cm stem and 50-cm arms (width, nine cm; height, five cm). Further details of the experimental abalones are presented in Data S1. The experiments were conducted as follows.

   **Step 1:** To form the mucus trail on the surface of the y-maze, one arm of the y-maze was closed using a plastic plate, thus restricting where marker individuals were allowed to move (right arm: $n = 38$, left arm: $n = 37$). The marker individual was released at the end of the stem in the daytime and usually began moving within one hour, eventually reaching the end of the y-maze arm after which is was removed. The plastic plate was then removed to open both arms of the y-maze. If the marker individual remained at the end of the stem for over a day they were replaced with a new marker individual.

   **Step 2:** The subject individual was released at the end of the stem and its direction of creeping was observed. This step commenced at 18:00 h because *H. discus hannai* individuals are active after sunset. Followers often reached the end of the arms within several minutes. If a follower remained at the end of the stem for over 30 min, its direction of creeping was observed using a video camera (DVSA10FHDIR; Kenko Tokina Corporation, Tokyo, Japan) until 6:00 h. If a follower individual remained at the end of the stem for over a day,

the experiment was stopped. After finishing each experiment, the walls and floor of the y-maze were scrubbed clean using a spongy, prior to beginning subsequent experiments.

The effects of marker-tracker combinations and season on MTF rates were tested using a generalized linear model (binomial distribution) with likelihood ratio tests using the "Anova" function in the "car" package for R v3.5.2. We also used a binomial test to determine whether MTF was actually being observed, without dividing the subjects into experimental groups.

## Mucus trail following as a mate-finding strategy

To examine whether MTF behavior reduces the distance between members of the opposite sex, an IBM consistent with the behavior of *H. discus hannai* was constructed (see also Fig. S1) using NetLogo v5.3.1. The script is available on GitHub with the following file name: File S1 (https://github.com/YMatsumoto5536/PeerJ-netlogo).

In this model, female MTF abalones (i.e., mutant females), and non-MTF female and male abalones (wild females and wild males) were generated. The IBM compared the distances between wild males and mutant females and between the wild males and wild females. The comparison revealed whether MTF is effective for reducing the distances between the members of the opposite sex from the aspect of female benefit.

### *Spatial units and time scale of the model*

The density of *H. discus hannai* varies by year and location within the Iwate Prefecture; the observed density ranges from 0.56 to 3.82 individuals $m^{-2}$ (*Ohmura et al., 2015*), with a range of 1.0–1.5 individuals $m^{-2}$ occurring most frequently. We therefore implemented an abalone density of 1.0 individual $m^{-2}$ in the model. Spatial units were 10 cm $\times$ 10 cm grid cells and the model consisted of a 3 $m^2$ area (i.e., 30 $\times$ 10 grid cells). The boundaries of the model world were reflective: when individuals reached a boundary, they turned 90° and resumed movement in the new direction. Each time step represented one day and the models were run for 30 steps (i.e., 30 days).

### *Movement algorithm*

Mutant females searched the grid for a mucus trail within a 10 cm search radius. They next decide whether to move to a grid cell with mucus or an empty grid cell according to their individual mucus following rate parameter (see below). The individual moving distance for each day was generated from an exponential distribution with mean of 48.76 cm, which is consistent with observed behavior in the holding tanks (Data S2). The individual MTF rate was generated from a binomial distribution (trial number = 75, number of arms with mucus = 60, estimated following rate = 0.8). This following rate is consistent with the observed behavior in the tank experiments (Fig. 1). Model abalones followed mucus trails irrespective of the sex combination, consistent with the only small sex combination effects observed in the tank experiment (Fig. 1). Although the wild females and wild males moved the same distance as the mutant females, their direction of movement was random. All individuals left mucus trails before moving to the next grid cell. Studies have suggested that the mucus trail of gastropods is degraded by bacteria within a single day (*Herndl & Peduzzi, 1989*; *Peduzzi & Herndl, 1991*). Therefore, we set the functional persistence period of the

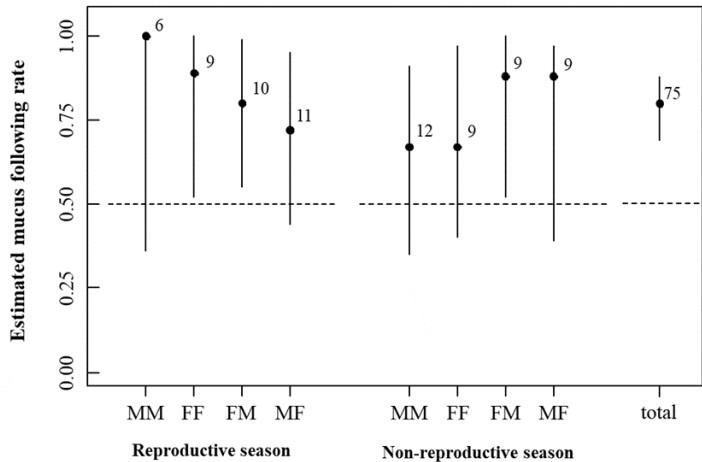

**Figure 1** **Estimation of the mucus following rate with eight marker-tracker sex combinations in the reproductive and non-reproductive seasons.** Error bars indicate 95% confidence intervals. Numbers represent sample sizes. Abbreviations for sex combinations: MM, male follows male; FF, female follows female; FM, female follows male; MF, male follows female.

mucus to one day in the IBM. These processes were repeated until the day's determined locomotion distance had been reached. At the end of the 30 day test period the distances between opposite sex individual was measured.

***Measuring the distance between opposite sex members***

Differences in the distances between mutant females and wild females were compared using a generalized linear mixed model (Gaussian distribution) using the "lmer" function in the "lme4" package for R v3.5.2. To conduct paired comparisons within simulations, the simulation number was treated as a random-intercept in the model. The effects of locomotion type on the nearest neighbor distance were tested with likelihood ratio tests using the "Anova" function in the "car" package.

## Evolutionary model of MTF behavior

To examine whether abalones evolved the ability to follow mucus trails to reduce distances between members of the opposite sex, we constructed simple population genetic models for a diploid population undergoing nonoverlapping, discrete generations (see also Fig. S2). It should be noted, however, that *H. discus hannai* undergoes repeated spawning over several years. The script is available on GitHub with the following file name: File S2 (https://github.com/YMatsumoto5536/PeerJ-netlogo).

***Dynamics of allele frequency***

In MTF model, the MTF mutant type and non-MTF wild type were evaluated for a single autosomal locus with two alleles, dominant (*D*) and recessive (*d*). *DD* and *Dd* individuals both followed the mucus trail (to the same extent), while *dd* individuals did not. The behavioral rules of mutant type and wild-type were otherwise the same as in the model described above. After the model was run for 30 days the nearest neighbor distances

between members of the opposite sex were measured. Each female fertilized its eggs with the nearest males. The number of eggs was determined using the following equation: the number of fertilized eggs $= 88.31^{-0.32 \times \text{distance}}$, as described by *Babcock & Keesing (1999)*. The rate of mutant type and wild-type in the next generation ($n+1$) with a population size of 10,000 was determined based on the ratio of the number fertilized eggs of each type in the current generation ($n$). This cycle was repeated for 2,000 generations and the frequency of each allele was assessed.

### Population size

The effective real world population size ($N_e$) of *H. discus hannai* is not clear. We thus set the population size to 10,000 because beneficial traits are removed by random genetic drift processes when the population size is small. A control model in which *DD* and *Dd* individuals did not follow mucus trail was constructed to confirm that the random genetic drift had little effect on the allele frequency in the MTF model. At the start of each simulation, 9980 individuals with *dd* (wild type) and 20 individuals *Dd* (mutant) were generated, assigned with a sex ratio of 1:1. Individuals were distributed randomly across a 10,000 m$^2$ area (1,000 $\times$ 1,000 grid cells).

## RESULTS

### Following the mucus trail of another individual

The MTF rates observed in each experiment are shown in Fig. 1. The marker and follower combinations (likelihood-ratio test, $df = 4$, $\chi^2 = 2.68$, $p = 0.61$), season ($df = 1$, $\chi^2 = 0.026$, $p = 0.87$), and their interaction term ($df = 3$, $\chi^2 = 0.36$, $p = 0.95$) did not affect the MTF rate. Abalones chose the arm with a mucus trail more frequently ($n = 60$) than the arm without a mucus trail ($n = 15$) when experimental groups were pooled (binomial test, $p = 1.588$ e$-07$).

### MTF reduces the distance between opposite sex individuals

The distance between wild males and mutant females (mean $\pm$ SD $= 10.83 \pm 6.57$ cm ) was significantly less than that between wild males and wild females ($11.08 \pm 6.61$ cm; $df = 1$, $F = 6.90$, $p = 0.009$); however, the estimation difference in distance was only 2.4 ($\pm$SD $= 0.8$) cm.

### Evolutionary role of reducing opposite-sex distances in MTF

As the frequency of *D* alleles did not increase over the generations in the control model (Fig. 2A), the increasing frequency of *D* alleles in the trail following model was caused by an increased fertilization rate rather than random genetic drift. In the MTF model, the frequency of *D* alleles (Fig. 2B, see also Data S3) increased over the generations 53 of the 100 simulations. Complete disappearance of the *D* allele occurred within 314 generations (average $\pm$ SD $= 88 \pm 69$ generations; range $= 13$–$314$) in the MTF model.

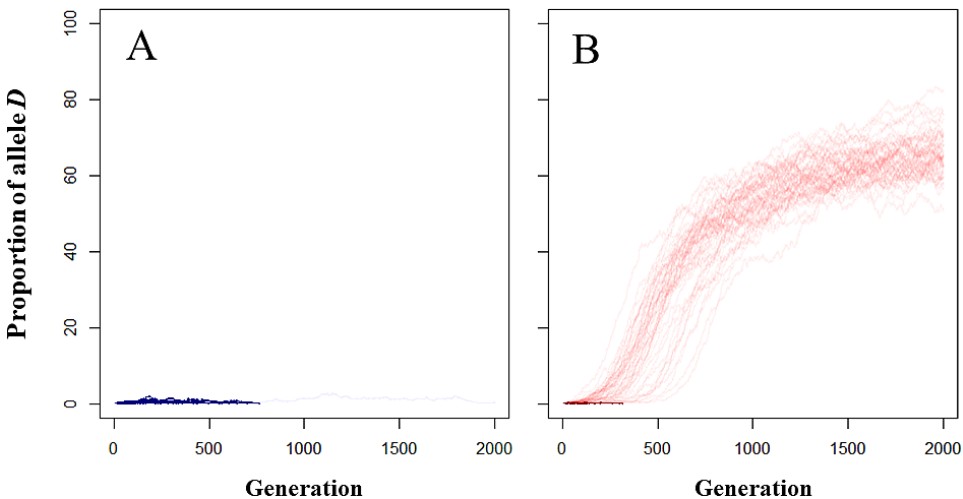

**Figure 2** **Dynamics of frequency of the *D* allele in the mucus following model.** (A) Control model; (B) MTF model. Light- and dark- colored lines correspond to simulations in which the *D* allele increased and simulations in which the *D* allele disappeared.

## DISCUSSION

### The MTF trait

Our experiments strongly suggest that the presence of a non-self mucus trail affects the directional decision making in *H. discus hannai*. Mucus trails of *H. discus hannai* are known to attract their larvae (*Roberts, 2001*); thus, it is not surprising that adult individuals also react to conspecific mucus trails. The tentacles of *H. asinina* react to protein extracted from mucus trails *in vitro* (*Kuanpradit et al., 2012*), further supporting the hypothesis that *H. discus hannai* senses mucus trails using their tentacles.

We found that *H. discus hannai* followed mucus trails irrespective of the marker's sex or season. As the experimental design did not allow for the selection of mucus from different sexes simultaneously, we cannot deny the possibility that *H. discus hannai* might prefer mucus from the opposite sex and sex-related differences in preference for mucus, although sex-related differences in mucus composition have not been found in *H. asinina* (*Kuanpradit et al., 2012*). Although the preference for mucus from the opposite sex in *H. discus hannai* was not detected in this study, there remains the possibility that mucus-trail-following shorten the distance between opposite sexes more efficiently than creeping without mucus-trail-following.

The factors that affect decision making about MTF should be investigated in the future, and not only the trail following rate. For instance, the trail following rate of *Nodilittorina unifasciata* increases when emersed during low tide (*Chapman, 1998*). Starvation could also affect MTF behavior because some gastropod species ingest mucus as an energy source (*Davies & Beckwith, 1999*; *Hutchinson et al., 2007*; *Ng et al., 2013*). Conversely, animals avoiding the trail of other animals may acquire more food than those that follow others.

## MTF reduces distances between members of the opposite sex

Although some studies have suggested that MTF increases the efficiency of mate searching (*Ng et al., 2013*), no study has produced a definitive demonstration. Previous investigations have focused on the rate of MTF rather than the function of this behavior. In this study, we used an IBM to show that the mean distance between opposite sex individuals was significantly reduced by MTF. However, the distance between males and mutant females was greater than that between males and wild females in 4,907 out of 10,000 simulations. Additionally, distances can be reduced when mutant females follow the mucus of wild males: in such cases the fertilization rate (i.e., reproductive success) of wild males increase even though they do not themselves exhibit MTF behavior.

Locomotion patterns might have affected the distances between males and females in ways other than MTF in our model. The random number generated from the probability distribution for the locomotion distance in *H. discus hannai* equated to lévy flight (Fig. S3), which is an effective searching behavior when the target is sparsely distributed (*Sims et al., 2012*). Observed locomotion pattern in another *Haliotis* species was found to involve lévy flight in the wild (*Strain, Johnson & Thomson, 2013*); therefore, the natural locomotion pattern of *H. discus hannai* might also involve lévy flight. The current study focused on MTF as a method of reducing distances between members of the opposite sex; however, other behaviors such as locomotion patterns should also be investigated in the future.

## Effect of increased fertilization rate on evolution of MTF behavior

The *D* allele increased within the population in the MTF model in just over half of the simulations (53/100 simulations). This result indicates mate searching (resulting in increased fertilization rates) could be the evolutionary driving force behind MTF in our model. However, the *D* allele disappeared from the population in 47 of the 100 simulations, possibly suggesting that the resulting increase in fertilization rate as parameterized is not an evolutionarily stable driving force in MTF. In general, changes in allele frequency depends on the product of the average fitness of the phenotype and the frequency of the allele within the population (*Kawata, 1989*). When the fertilization rate of the mutant is lower than or almost the same as that of the wild individual, the *D* allele disappears or does not increase at the beginning of simulations when the allele frequency is low. As described above, the observed distance between males and mutant females was greater than that of wild females in almost half of the simulations (4,907/10,000 simulations): this might explain why the *D* allele did not always increase early in the simulation. The wild type *d* allele was not perfectly removed from the population in all simulations (Fig. 2), possibly because the fitness of followed wild individuals was increased by the MTF behavior of mutant individuals without incurring any costs. The *d* allele might also persist in heterozygous *Dd* individuals, which express the same degree of MTF behavior as homozygous *DD* individuals.

As allele *D* did not increase in about half of the simulations, other benefits in addition to increased fertilization rates could be the evolutionary driving force of MTF in the real world. For example, MTF reduces the energy cost for adhesion and locomotion; gastropods require energy to secrete a mucus trail to adhere and move across the substrate (*Davies & Hawkins, 1998*). An individual following a mucus trail can decrease the amount

of mucus being secreted (*Davies & Blackwell, 2007*; *Hutchinson et al., 2007*), thus saving energy. *Haliotis discus hannai* exhibits MTF in the non-reproductive season, suggesting a possible energetic saving. Unlike the mate-search function, energy savings would accrue continuously.

In the future, the following hypotheses should be tested as factors driving the evolution of MTF as a mate searching strategy. First is that the functional persistence period of trail mucus affects the encounter rate by others. The model's persistence period of mucus trail was set following observational studies showing that mucus is degraded by bacteria within a day (*Herndl & Peduzzi, 1989*; *Peduzzi & Herndl, 1991*). However, the trail mucus in *H. asinina* could potentially remain on the substrate and be functional for as long as a week (cited as an unpublished observation in *Kuanpradit et al. (2012)*. If the mucus has such a long functional period, abalones could encounter the trails of others much more frequently, which could enhance the distance-reducing effect between opposite sexes. The second hypothesis is the effect of abalone density on the efficiency of MTF behavior. It is possible that abalones are less likely to find the mucus trail of other individuals at low population density, and vice versa. Although the density of abalones was set 1.0 individuals $m^{-2}$ in the model, the density varies by year in wild populations (*Ohmura et al., 2015*). Therefore, the efficiency of MTF behavior could vary by year in the real world, and this should be a focus for future work. In this study, expression of MTF behavior is determined by allele $D$ and $d$, with nonoverlapping generations; however, the actual relationship between MTF and allele is not clear, and *H. discus hannai* undergoes repeated spawning for several years. An improved model to address these problems is needed to understand the evolution of MTF behavior.

## CONCLUSIONS

This study indicates that trail mucus affects the direction of movement in *H. discus hannai*. Additionally, the possibility that MTF behavior has evolved as a mechanism to increase the fertilization rate was indicated by the IBM. Our simple genetic models indicate that increased fertilization rates are not an evolutionarily stable driving force of MTF because the $D$ allele, which determines the behavior, was 47 out of 100 simulated populations. Thus, an additional and possibly additive mechanism is required to explain the fixity of the trait in natural populations. *H. discus hannai* individuals follow conspecific mucus trails irrespective of the reproductive season or sexes, supporting the possibility of the presence of other evolutionary driving forces acting on the trait.

Estimating the extent of aggregation and nearest neighbor distance is important for the population management of *Haliotis* species (*Button, 2008*) because these factors affect egg fertilization rate (*Babcock & Keesing, 1999*). Aggregation patterns in wild populations of *Nodilittorina unifasciata* can be modeled using a process in which individuals follow the mucus trails of others, rather than a process in which individuals remain in a preferred spot on a rock (*Stafford, Davies & Williams, 2007*). An IBM based on MTF behavior might be helpful for predicting aggregations of *H. discus hannai,* however, location preferences based on the environmental characteristics might similarly affect the aggregation. Location

preferences and benefits based on environmental characteristics should be studied in wild populations to help parameterize future iterations of IBMs to assist in practical population management.

### Funding

This work was supported by the JSPS KAKENHI (15K18734) and Tohoku Ecosystem Associated Marine Sciences. The funders had no role in study design, data collection and analysis, decision to publish, or preparation of the manuscript.

### Grant Disclosures

The following grant information was disclosed by the author:
JSPS KAKENHI: 15K18734.
Tohoku Ecosystem Associated Marine Sciences.

### Competing Interests

The author declares there are no competing interests.

### Author Contributions

- Yukio Matsumoto conceived and designed the experiments, performed the experiments, analyzed the data, prepared figures and/or tables, authored or reviewed drafts of the paper, and approved the final draft.

### Data Availability

The raw measurements are available in the Supplemental Files.
The models are available in GitHub
(https://github.com/YMatsumoto5536/PeerJ-netlogo): File S1 and S2. The files are written in NetLogo 5.3.1.(https://ccl.northwestern.edu/netlogo/).
The model procedures are available in Figs. S1 and S2.

### Supplemental Information

Supplemental information for this article can be found online at http://dx.doi.org/10.7717/peerj.8710#supplemental-information.

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
