# Peer review of "Effects of mucus trail following on the distance between individuals of opposite sex and its influence on the evolution of the trait in the Ezo abalone Haliotis discus hannai"

_PeerJ, doi:10.7717/peerj.8710_

## Round 0.1 · original submission · Major Revisions

I have heard back from three reviewers. Each offer different but constructive comments to help you improve your submission. Specifically, I agree with the reviewers that the English needs thorough editing, and also with the third reviewer, who asks questions about your choice(s) of evolutionary model and would like to see a more well-developed Discussion. Please also do not forget to check the annotated manuscript submitted by the first reviewer.

I look forward to seeing a revised version.

Reviewer 1 ·

Basic reporting

The use of the English language will still require considerable attention. There are quite some odd wordings and use of grammar.
Some supplemental data were lacking. I recommend Table 1 is expressed in graphical form for easier assessment.
Other specific comments are included in the attached file.

Experimental design

“Materials & Methods” section is highly difficult to follow and needs rephrasing. Specific comments are included in the attached file.

Validity of the findings

The author should also consider whether abalone follow mucus or not depends on environmental factors. Specific comments are included in the attached file.

Additional comments

This is an interesting paper dealing with evolutionary benefits of mucus trail following in abalone species using observation of tank experiments, individual-based models, and population genetic models. While I am briefly familiar with the behavioral ecology of abalone species, I am not an expert in genetic models and cannot adequately comment on the technical aspects of data analysis. However this paper would be a contribution on the behavioral and reproductive ecology of important abalone species.

Annotated reviews are not available for download in order to protect the identity of reviewers who chose to remain anonymous.

Reviewer 2 ·

Basic reporting

A bit disjointed and unclear. The findings are interesting, but not well presented. At least one of the references is a bit misleading, as mentioned in general comments. Seems to be a few bits of evidence thrown in randomly without really explaining their relevance or significance.

Experimental design

It is not explained why a model is used rather than the real animal assay conducted in the trail-following experiment, and there is inconsistent use of terminology. Ie. Encounters or distance between?

Validity of the findings

I am not qualified to comment on the validity or reliability of the model-based experiment.

Additional comments

Line 33 – was this a significant difference?
40 – externally fertilized
43 – opposite sexes
53 – through olfaction
55 – the opposite sex
57 – following mucus trails
60 – with the opposite sex
61 – Are they testing distance between or encounter rate? Those two things don’t seem to be the same to me…
64 – this is not worded very clearly
71 – and in the field
73 – this doesn’t make much sense to me
77 – conditions
79- Why is Fig. S1 not in the results section, or was it performed in a previous study?
82 – In this ONE study, it only implies this, and only in relation to food. A big reach to make this statement I think.
86 – Should stipulate that this was using a model, to differentiate between the previous investigation using live animals
94 – These species?
111- starting point – what kind of camera?
128 – this is not clear, needs rewording
139 – Not S4, think it should be Data S2
143 – Can’t see how this relates to what I think is Figure 1
185 – this is confusing – no Supplemental Data S3 received in files
187 – Results (the?)
194 – Would be good to have absolute numbers here. Ie how many followed trail?
223 – Grammatically incorrect, needs rewording
237 – Opposite sex members of wild males? – doesn’t make sense
270 – trail
283 – not the only evolutionary driving force
284 – was not were

Reviewer 3 ·

Basic reporting

The manuscript “The effects of trail following on the distance between opposite sex and the effect of its benefit on the evolution of trail following in Pacific abalone Haliotis discus hannai” tries to determine the importance of mucus trail following behaviour as an adaptive trait. This is a very complex eco-evolutionary issue of broad biological relevance.

The English should be improved, the ideas are not clear and in some paragraphs there are several contradictions. This makes it hard to access the manuscript. For example, in lines 63-65, 72-73, 77-79, 255-261 the current phrasing makes it very difficult to follow the ideas. I recommend that the author have a native English speaker review the text.

Although it is difficult to follow, the background is sound enough to have an idea of the main question. The structure of the manuscript follows a traditional format. However, the subdivisions of the methods make it difficult to read and the subtitles are extremely wordy without providing relevant information, especially the part describing the models (Lines 133-159).

I think the results are enough to test the hypothesis proposed, but the evolutionary model has several problems (see below). The figure in the main text could be the most important result, but some modifications to the model should be done (see my comment below). The table could be in the supplementary material, substituted for a figure showing the same results.

Experimental design

The author used mainly the correct statistical and experimental approaches. These allowed to them to establish the prevalence of trail following behaviour in the species (although this has been reported in several species of molluscs) and that this behaviour reduces the distance between individual of the same species. However, these methods were not enough to determine the evolutionary consequences of this behaviour, which would be the most broadly relevant result in the manuscript.

It is very weird that the two models (trail following and drift) show very similar results. More specifically, the high number of simulations where the allele D got lost in both models is huge; this, and the fact that both models reach a maximum of 90% frequency of the D allele around the same number of generations, makes me believe something is wrong with the simulations. I can understand how the advantage of Dd in the trail following model prevents a fixation of D, but why does this fixation never occur in the neutral model? Perhaps another problem is the simulated density; most species of abalone are distributed in patches with natural densities over one individual per m2 (Goodsell et al 2006). As the author pointed out, the density is more relevant for evolutionary processes than a population census size without geographic context. This is especially true in species with low mobility such are the abalone. I suggest including models with slightly higher abalone densities to better represent the wild populations.

Validity of the findings

As I mentioned before, I think the results are well supported, except for the evolutionary consequence of trail following behaviour. I personally found the discussion very limited, underexploring the results. For example, I suggest more discussion about the possible problems in the evolutionary model.

---

## Round 0.2 · Minor Revisions

I have heard back from two reviewers, both of whom also reviewed the first draft of your work. One reviewer only offers minor comments, but the other reviewer has two main critiques. One is that the neutral model is now not included, and asks for it to be re-placed back into the work. It might be better to do so unless you have a compelling scientific reason for its removal, which should be detailed in your responses.

The other critique is that the paper is still awkwardly written in places. While I can appreciate the difference of writing in another language, I agree with the reviewer that there are many instances in which the language seems stilted and unnatural. For example, you often use the phrase "mucous trail following" with no quotes or hyphens, but this phrase as written seems more like a verb phrase than the subject of your study. I suggest finding a native English speaking colleague (and not the editing service you have used until now) and having them go over it once very thoroughly with you. Issues such as singular or plural, and phrases and terms to connect sentences, will all make you paper much easier to read and follow.

Please contact me if you have any questions, and I look forward to seeing your revised version.

Reviewer 1 ·

Basic reporting

No comment.

Experimental design

No comment.

Validity of the findings

No comment.

Additional comments

The paper has been improved since the prior submission. Please see my minor comments as below.

Line 29: “arm” -> “y-maze arm”

Line 47: “higher rates of fertilization rate” -> “higher rates of fertilization?”

Lines 97-98: The author should show how to observe the sex of non-reproductive individuals.

Lines 105-106 and 109-110: I notice that follower abalone were placed on mucus-attached substrates at the beginning of trials, so the author examined the trait of abalone avoiding mucus, rather than that of abalone attracted to mucus? I think that the author should examine whether abalone are attracted to mucus and they follow it, and this point should be discussed.

Lines 154-155: Supplemental dada S2: Perhaps the measuring moving distance was conducted under a lower temperature than the Y-maze experiments?

Lines 199-200: “The combination of marker and follower combinations” -> “The marker and follower combinations?”

Line 312: “species”; not italic.

Reviewer 3 ·

Basic reporting

The authors improve the manuscript in this second round, however I still have difficulties to follow the ideas. Some paragraphs are repetitive (e.g. Lines 146-147 and Lines 153-155), and some ideas are not clear (e.g Lines 57-70).

In terms of format, I think the authors did not change enough. I think the methods are subdivided too much, with the subtitles still extremely wordy without providing
relevant information. The simulation model is intrinsically complex, and this style of writing makes it more difficult for the reader to understand.

The previous version has enough results to test the hypothesis proposed, but this version has removed the neutral evolutionary model. These simulations are important to distinguish between a random effect and a selection effect on the allele frequency due to mucus following behaviour.

Experimental design

The statistical and experimental approaches are mainly correct. The methods are related to the questions but there are still problems with the simulation model. The authors removed the neutral model (random), which is fundamental to establish the evolutionary effect of the trail following behaviour.

Validity of the findings

The results are clear about the prevalence of trail following behaviour and its effect on the distance between individuals. However, the evolutionary consequences of this behaviour are still unclear. The authors modified the model as requested. But still, there is not clear evidence for the effect of this behaviour on the reproductive success of the species. I think this is related with the bad writing style of the manuscript but also the removal of the neutral model results.

---

## Round 0.3 · Minor Revisions

Scientifically, this manuscript is ready to be accepted. Still, there are some mistakes in your edits, and the references are also inconsistently formatted, so I am sending this back to you for one more round of editing. Please see the attached MSWord file as well.

---

## Round 0.4 · accepted · Accept

Thank you for your hard work; I am more than happy to move this into production and look forward to seeing your final published paper!